# Dead Feature Counts in Sparse Autoencoders Predict Underlying Deep Q Networks' Effectiveness

**Coleman DuPlessie**
Massachusetts Institute of Technology
Cambridge, MA 02139
`cdupe@mit.edu`

## Abstract

Sparse autoencoders (SAEs) are machine learning models that can be used to express the inner workings of certain other models as human-interpretable features. While sparse autoencoders work well when applied to language models, there has been little research that investigates the extent to which they generalize to other applications of machine learning. This work investigates the application of SAEs to a deep Q network trained to complete a simple task. We find that, although SAEs tend to perform well and find a number of human-interpretable features, they contain a large number of "dead features" that never activate, which suggests that more research is necessary to adapt SAEs to the unique tasks reinforcement learning models solve. In particular, we note that the most effective deep Q networks trained to complete a task tend to result in sparse autoencoders with a consistent quantity of dead features. This suggests that these sparse autoencoders may in some sense be capturing the "optimal" or "true" number of features needed to solve the toy problem we study, and the high number of dead features may simply imply that additional live features past a certain quantity are unhelpful.

## 1 Introduction

Recently, there has been great progress in the field of mechanistic interpretability, which studies methods used to make the decision-making processes of trained machine learning models understandable to humans. The vast majority of this work has centered around transformers, which provide an ideal testing ground for three reasons: first, their inputs and outputs (natural language) are very easily understood and manipulated by humans; second, natural language tends to be extremely conceptually sparse (i.e. for any concept, the vast majority of written text is unrelated to that concept); and third, there are immediate practical uses for interpretability in transformers today (e.g. being used as a method to fine-tune the outputs of Large Language Models (LLMs) without requiring large amounts of human feedback [12]). However, other model architectures could also benefit significantly from the application of recent advances in interpretability.

This paper focuses on the use of sparse autoencoders (SAEs). SAEs have been successfully applied to transformers to decompose their activations into human-interpretable features [4], which can then be manipulated to change the transformer's output in meaningful, interpretable ways [12].

In this paper, a variety of sparse autoencoders are trained on the activations of deep Q networks (DQNs). Many, though not all, features of these SAEs appear interpretable. Additionally, there are a significantly elevated number of dead features when compared to models trained on transformers, and, surprisingly, the number of dead features in a SAE is a statistically-significant predictor of the underlying DQN's performance on our toy task, with SAEs with a moderate number of dead features correlating with the best performance (see Figure 1).

Preprint.

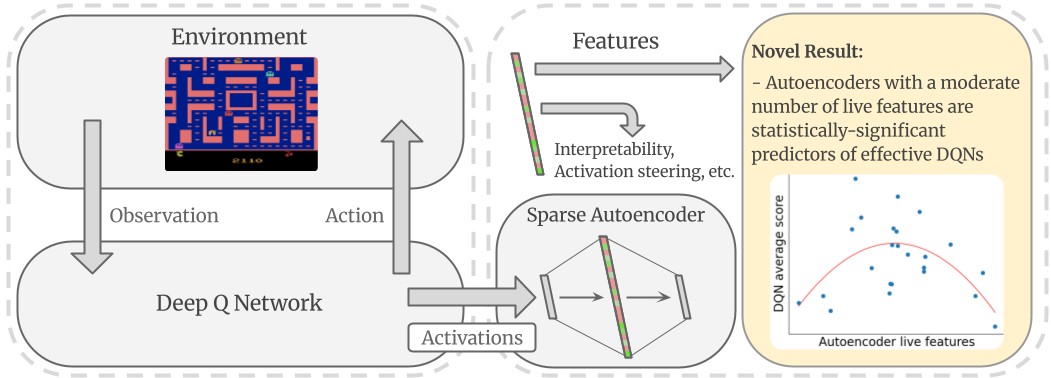

Figure 1: We find a relation between the number of live features in a sparse autoencoder and the average performance of the deep Q network it is trained on

## 1.1 Sparse Autoencoders

Sparse autoencoders are a variety of autoencoder useful for taking features out of superposition. Superposition refers to the theory (proven to exist in toy models, and conjectured to hold in many large models) that "features," independent concepts represented by a machine learning model, are not actually represented independently, one in each neuron. Instead, features are each represented by a linear combination of neuron activations such that the set of features forms an overcomplete basis for the activation space of the model. This means that the model is able to represent more features than it has neurons, at the cost of reduced performance caused by independent features interfering with each other. Because of the risk of feature interference, superposition is most common when features are very sparse and any given feature is inactive on (i.e. irrelevant to) the vast majority of input data [7].

If a model internally represents features in superposition, this implies that its neuron activations will not be interpretable by themselves, since each neuron is then representing a linear combination of disparate, unrelated features. This is the fundamental problem of mechanistic interpretability, and necessitates some method of taking features out of superposition.

This is achieved by training a sparse autoencoder, a network trained to make its output equal to its input, on the neuron activations of an underlying model. Non-sparse autoencoders are often used to learn efficient codings of arbitrary data, by making their hidden size smaller than their input and output size. SAEs, on the other hand, have hidden sizes substantially larger than their input size (at least 2 times as large, but more commonly anywhere from 4 to 256 times as large). SAEs are trained on some part of the internal state (generally, the activations of one particular layer) of a pre-existing model in order to interpret the model's decision-making process. In order to ensure that SAEs pull features out of superposition (thus rendering them interpretable), SAEs have a mechanism to encourage sparsity (thereby discouraging superposition, since features are sparse, but sets of many features in superposition are generally quite dense). There are two primary mechanisms used for this purpose: sparsity penalties and k-sparse autoencoders. This research uses k-sparse autoencoders.

k-sparse autoencoders allow a fixed number of neurons to fire on any given input. They do this by replacing a ReLU function or other nonlinear activation function with the TopK activation function, which allows the $k$ highest-activating neurons to remain unchanged (similar to all positive neurons with a ReLU activation function) while setting all other neuron activations to zero.

This approach has multiple benefits. It is extremely easy to tune the L0 norm (i.e. average feature sparsity) of k-sparse autoencoders, since their L0 norm is forced to be equal to $k$. Due to this enforced sparsity, the activations of a SAE's hidden layer are referred to as "features" (though some of these features may not be interpretable). k-sparse autoencoders also avoid problems like shrinkage associated with autoencoders that use sparsity penalties and therefore tend to have higher accuracy [8].

## 2   Related Work

Previous work in this area has primarily focused on the application of sparse autoencoders to large language models, built around the foundation established by Cunningham et al. [6], who showed that sparse autoencoders were fundamentally able to find features in language models. This approach was later refined by other work, such as Bricken et al. [4] and Templeton et al. [12], which introduced performance-enhancing techniques such as resampling and showed that sparse autoencoders scale well as underlying models grow larger.

Makhzani and Frey [8] introduced the k-sparse autoencoder as an alternative to L1 loss penalties, which were previously ubiquitous in regulating overall sparsity. Various other techniques have been proposed to improve model sparsity, such as Rajamanoharan et al. [9], which introduced the gated SAE, or Rajamanoharan et al. [10], which found benefits from using jump ReLU units in SAEs. These three techniques are mutually exclusive as they all depend on the model's activation function: only the former was pursued in this paper.

The use of sparse autoencoders was expanded to image models by Surkov et al. [11], who showed that SAEs could reliably find interpretable features in text-to-image models. Researchers such as Abdulaal et al. [1] have been able to apply sparse autoencoders trained on vision-language models to achieve superior results to fine-tuning on practical tasks such as generating radiology reports.

There has been very little previous research into the application of SAEs to reinforcement learning models. Annasamy and Sycara [2] developed architectures for DQNs designed to aid in interpretability, but, being written before Cunningham et al. [6], did not discuss sparse autoencoders. A preprint by Yin et al. [13] released in November 2024 showed that SAEs compare favorably with other reinforcement-learning-based alignment methods, such as Direct Preference Optimization (DPO) or Reinforcement Learning from Human Feedback (RLHF), when applied to large language models. However, it focused exclusively on the alignment of pretrained large language models, rather than models trained with reinforcement learning "from scratch."

This paper is the first to apply sparse autoencoders to Deep Q Networks, a more traditional reinforcement learning model architecture. Additionally, unlike previous work, we quantitatively investigate the relationship between dead features in sparse autoencoders and the performance of the underlying models.

## 3   Methods

In this paper, SAEs are trained to allow human interpretation of the features of a small DQN, which is itself trained on a simple reinforcement learning task. This paper uses the game Ms. Pacman for the Atari 2600 as a reinforcement learning environment.

### 3.1   Environment

This research uses OpenAI Gym, a standardized API for reinforcement learning [5]. The environment we use is the Ms. Pacman game, made available through the Arcade Learning Environment, which is available under a GPL-2.0 license[3]. When a reinforcement learning agent is being trained, the model's reward is equal to its score, and its input is a live feed of what the player sees. The live feed is represented as a tensor of size 3x210x160.[1] The agent must output one of nine possible directions to push the game's virtual "joystick" (the 4 cardinal directions, 4 diagonals, and the center). Additionally, the model is only queried every fifth frame. This is a common feature of many video-game-based reinforcement learning environments: since the optimal move is likely to remain the same between two adjacent frames, it is minimally helpful to repeatedly query the model on near-identical questions. Instead, it is more efficient to train for more games with a lower frame rate.

Ms. Pacman (see Figure 2) is a useful environment for interpretability research because it is fully deterministic (enemies' movement is pseudorandom based on how the player moves) and relatively

---

[1]Due to the physical design of the original Atari 2600, the environment's vertical resolution is greater than its horizontal resolution, meaning that pixels are not square. Depictions of the model in this paper use square pixels, causing the model's input to appear taller than it is wide, despite the original game (and depictions of gameplay in this paper) having a 4:3 aspect ratio.

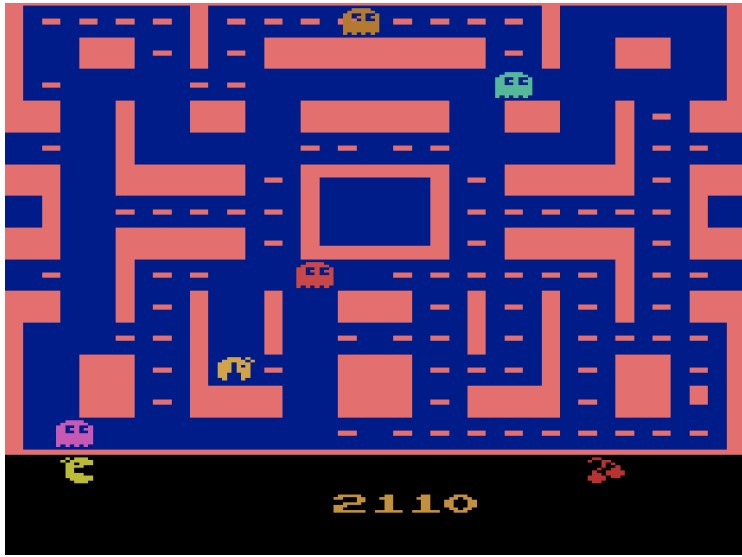

Figure 2: A game of Ms. Pacman being played by a trained DQN

simple while also very sparse (i.e. there are many features a model might want to track, but few of them are relevant at any given moment). This is important because SAEs are only useful due to superposition in the base model, and superposition tends to be stronger when features are sparser [7]. Another benefit of this environment is that it is highly interpretable to humans, and models' decisions (and mistakes) are easily understood, unlike other, more obscure or detailed reinforcement learning tasks.

## 3.2 Training

Training consisted of two phases: first, training a DQN to perform well in the environment at hand (Ms. Pacman), and next, training a sparse autoencoder to reconstruct the DQN's second-to-last layer's activations. In both cases, several different model architectures and hyperparameters were tested to maximize accuracy and, in the case of the SAE, interpretability. The most effective are discussed here.

### 3.2.1 Deep Q Network

Experimentation began by training a base model, a deep Q network, for 20,000 games. The DQN's architecture consisted of 3 progressively smaller convolutional layers (with one pooling layer) followed by 3 fully-connected layers (see Figure 3). All layers used a ReLU activation function.

The DQN is relatively small, with less than 880,000 parameters. This causes it to attain modest success with the task at hand (a much larger model would perform better), but it is large enough to contain interpretable features, while being small enough to only contain a moderate amount of such features, making it an ideal size for research applications. Prior work applying SAEs to large language models found that SAEs tend to become more interpretable, not less, as the base model grows larger [12]. This suggests that SAEs are likely to also scale well when applied to reinforcement learning models.

| Source of randomness | DQN train time | SAE train time | Test time |
|---|---|---|---|
| Random action | 3.75%–15% | 0% | 0% |
| Sticky action | 25% | 25% | 25% |

Table 1: Total frequency of random interventions in the DQN's behavior at train and test time.

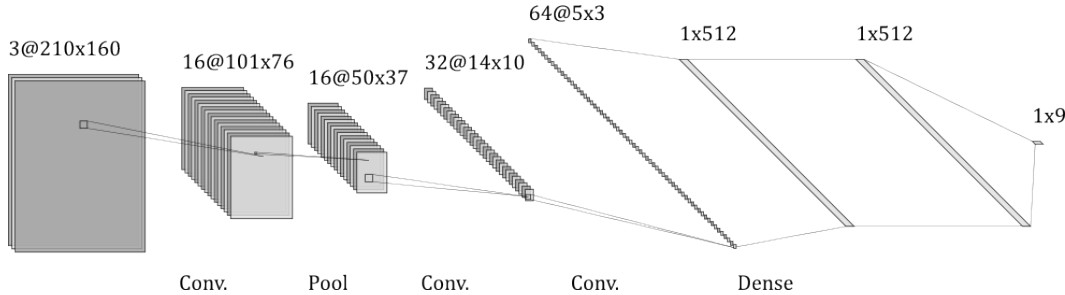

Figure 3: The architecture of the DQN used in this paper.

Notably, the DQN used in this paper has no recurrent layers or ability to act based on the past, meaning that it has no sense of time and makes decisions entirely based on the current frame. This is not an issue for model performance, because the environment (Ms. Pacman) displays all information that might be necessary to make the optimal decision each frame.

The DQN used in this paper was trained using the Adam optimizer, with a learning rate of 0.0005, batch size of 256, and $\beta_1, \beta_2 = 0.9, 0.999$. The DQN's gamma value is $0.98$ and its experience buffer is 50,000 samples long.

**Encouraging Randomness**    One of the benefits of using Ms. Pacman as an environment is that the game is pseudorandom: enemies' movements begin consistently but later depend unpredictably on the player's past actions (in OpenAI Gym's implementation, which this research uses, player input is the only such source of entropy for the game's pseudo-random number generator [3]). However, this threatens the DQN's ability to generalize: it is well within the abilities of even small reinforcement learning models to memorize the string of actions that best manipulates enemies' movement, and memorizing how to manipulate the enemies' actions replaces the process of making decisions based on the game state, meaning that the model's behavior no longer generalizes to other applications of reinforcement learning (since memorizing the one best sequence of actions is a degenerate case of reinforcement learning optimization).

To counteract this, randomness is added to the model's actions during the training process. Two methods are used to accomplish this, summarized in Table 1.

First, during training, there is some small chance $\epsilon$ that any given move is selected randomly instead of being chosen by the DQN. This ensures that the model eventually deviates from the solution it has mapped out up to this point. The value of $\epsilon$ can be tuned to strike a balance between the exploration of alternative strategies (when $\epsilon$ is large) and the exploitation and refining of the model's current strategy (when $\epsilon$ is small). In this research, $\epsilon$ begins at a value of 20% and decreases linearly throughout the first half of the training process to 5%, where it then remains for the second half of training.

This research also uses stochastic action stickiness: in addition to the varying chance that a random action is selected, there is also a constant, 25% chance that the model's previous action is selected.[2] Since the model is only queried every 5 frames to begin with, this means that there is a significant chance the environment goes 15, 20, or more frames without allowing the model to react to changes in the game state. This has two benefits: first, it encourages the model to "look ahead" at the decisions it is likely to make in a few frames and make them now if possible. If the player would be indifferent between two actions (e.g. pushing the joystick left or right while the player is in a vertical corridor), it

---

[2]This 25% chance of a repeated action is checked before the varying chance of a random action, meaning that the "true" chance of a random action ranges from 15% to 3.75%, not 20% to 5%.

is beneficial for the model to choose based on which action would be better if it is "sticky" for several frames (e.g. at the next intersection, would it be better to turn left or right?). Second, stochastic action stickiness provides another source of randomness: the model, through no fault of its own, has a chance of missing actions that it would have preferred to take. This ameliorates the problems and bad incentives caused by the environment's deterministic nature.

### 3.2.2 Sparse Autoencoder

After our DQNs have been trained, sparse autoencoders are trained on the activations of the DQNs' second-to-last layers. Each fully trained DQN plays 20,000 additional games, and SAEs are trained on DQNs' activations during these games.

During these additional games, there is still a 25% chance that the DQN's actions are sticky, but $\epsilon$, the chance of taking a completely random action, is reduced to zero. This is because the DQN's parameters are no longer being updated, and therefore there is no benefit gained from encouraging the DQN to explore all possible paths. Instead, the randomness produced by stochastic sticky actions is preferred, since it produces a distribution of games that varies more subtly. This allows the SAE to train on a diverse training set while ensuring that the training set still closely resembles the DQN's actual gameplay decisions.

The SAE's architecture itself is extremely simple: its input and output are each 512 neurons wide, and they are each connected to one hidden layer that is 2048 neurons wide. The hidden layer uses a top-$k$ activation function with $k = 50$. The hidden layer is 4 times larger than the input and output layers: this is a moderately small SAE, but still well within the range of sizes shown to be effective in other contexts (e.g. [4]).

Our SAEs also use a pre-encoder bias, in which an SAE's decoder layer's bias term is subtracted from the input to its encoder layer. This is motivated by previous work, such as Bricken et al. [4], which finds that pre-encoder biases improve model performance without harming interpretability.

## 4 Results

### 4.1 Deep Q Networks

24 DQNs, each with a corresponding SAE, were trained as described in Methods. Unsurprisingly, once fully trained, the DQNs vary significantly in performance, with models' average in-game scores (i.e. the average total reward attained over the course of an episode) ranging from 358 to 2411 points.

### 4.1.1 Interpretability

Upon inspection, most trained DQNs exhibit human-like strategies (e.g. seeking power-ups when in danger). However, without a SAE, there is no mechanism to draw out these strategies into concrete features that are connected to the model's weights, since neurons in our DQNs do not correspond one-to-one with features (generally, this is because there is no reason for features to be aligned with individual neurons and features are often stored in superposition [7]). Such a mechanism would make it possible to arbitrarily modify our DQNs' behavior in human-interpretable ways (e.g. no longer seek power-ups when in danger or always seek power-ups) without large side-effects on overall model quality.

Due to the small size of the DQNs used in this paper, even the best DQNs often make significant unforced errors such as failing to evade easily-avoidable enemies. Notably, these errors are marginally more common when the environment is in a state dissimilar to the DQN's training distribution (i.e. a state that is very uncommon for the model to arrive at by itself). This is expected, since the model has been trained to perform optimally with only 20,000 games, meaning that it has no experience acting in games that do not resemble these 20,000. Despite this, the model is often, but not always, able to perform well in radically out-of-distribution situations (e.g. if the model is not queried at all for the first 10 seconds of gameplay, leading to game states that are significantly different from any that were seen during training).

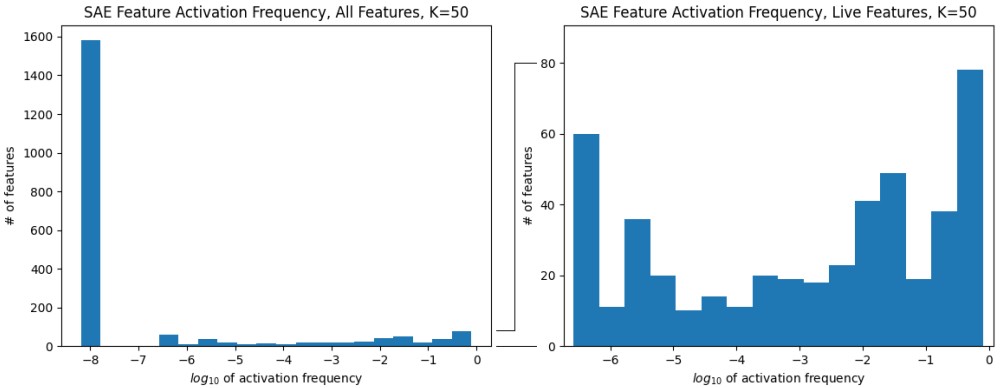

Figure 4: Left: Distribution of feature activation frequencies in one of our autoencoders. The tall bar to the left of all the others represents dead features that never activate. Right: the same distribution, with dead features cropped out.

## 4.2    Sparse Autoencoder Sparsity

We are able to measure sparsity of each of the 2,048 individual features in our sparse autoencoders by counting how frequently they activate across a sample of 2,500 games. This is valuable because, although the proper degree of sparsity does not necessarily guarantee interpretability, moderately sparse features have been shown to be more likely to be interpretable features than features in the so-called ultralow density cluster that activate on a nearly negligible fraction of all possible inputs [4].

Figure 4 shows the distribution of feature sparsities in one of the SAEs trained in this paper. Features that are extremely dense (in our case, activating on more than roughly 10% of all inputs) or extremely sparse (in our case, activating on less than roughly 0.1% of all inputs) are unlikely to be human-interpretable. The SAE trained here, although it does possess a number of highly dense features and ultralow density features, also contains a significant number of features of the proper sparsity, suggesting that these features are likely to be interpretable.

In addition to the moderate number of highly dense and highly sparse features, SAEs tend to also consistently contain a nontrivial number of dead features. A dead feature is defined as a feature that, regardless of the model's input, never has a nonzero activation. Measuring the number of dead features in a SAE with certainty is not possible without enumerating every possible input state, which is computationally infeasible. However, testing features on a large, but finite, set of input states is reasonably accurate (although it is likely to systematically overestimate the number of dead neurons in a SAE by a small amount: this method likely lists a moderate number of ultralow density features as dead, but this is not a significant problem because features that are sparse enough to be mistaken for dead features are also too sparse to possibly be interpretable).

As can be seen in Figure 4, approximately 70–85% of our trained SAEs' 2048 features are dead. This ratio remains fairly consistent across the 24 SAEs that we have trained (see Dead Features for a discussion of the variation of this number in more depth). This number of dead features is abnormally high, with previous research finding that dead features and those in the ultralow density cluster together can make up as much as 10–50% of a SAE's features in some contexts [4]. However, this does not inherently invalidate our SAEs' usefulness: the primary problem with a large number of dead features is that, since the average feature activation frequency is fixed (equal to $\frac{k}{2048}$), a high number of dead features will correspond to an increase in the average density of live features, which could have negative effects on their interpretability. Mitigating this, we can see in Figure 4 that a large fraction of the live features in our SAE have sparsities that are neither extremely high nor extremely low, which means they are likely to be interpretable.

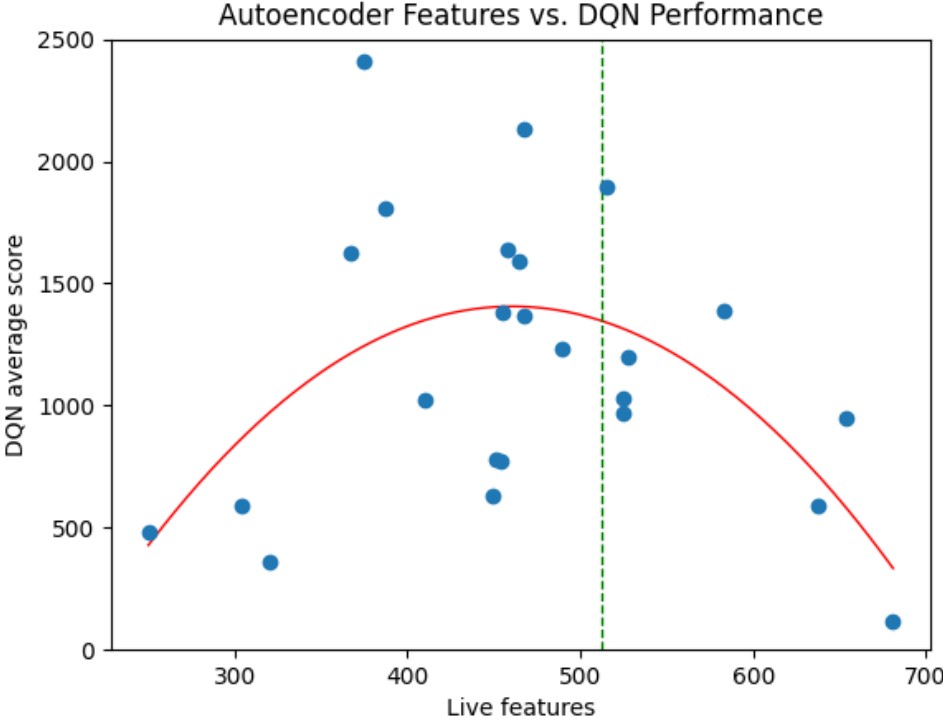

Figure 5: The relation between the number of live features found in a sparse autoencoder and the average performance of the corresponding DQN. DQNs that are more effective tend to create sparse autoencoders with a more consistent number of live features.

### 4.3 Sparse Autoencoder Interpretability

To determine the interpretability of individual features, the activations of several features were observed manually. This is in line with the prior state-of-the-art with SAEs, which uses human ratings to gauge the interpretability of features (e.g. [12]). The interpretability of features depends both on the input states that cause them to activate (if most states are similar and share a human-interpretable concept, the feature is likely interpretable) and the effect their activations have on the model's chosen action (if the effect is congruent with the input states, the feature is considered interpretable).

We find that most of the features that we would expect to be interpretable based on their sparsities are interpretable, but a large minority are not. This is consistent with previous work, where even in the best case, there are a significant number of uninterpretable features. Analyses of the interpretability of a few selected individual features can be found in Appendix A.

### 4.4 Dead Features

After calculating the number of live features in each sparse autoencoder by testing their behavior over 2,500 games, we are able to plot the number of live features against the underlying DQNs' performance.

Notably, we see that a majority of trained SAEs (17 out of 24) have less than 512 live features (denoted by the green line in Figure 5). This is significant because the layer of the underlying DQNs that our SAEs are trained on only contains 512 neurons, meaning that, despite containing interpretable features (unlike the underlying DQNs), our autoencoders often contain less than one live feature for every neuron in their respective DQN.

Intuitively, we would expect that SAEs with greater numbers of live features might denote higher-performing DQNs (since the existence of large numbers of live features appears to suggest a more complex reasoning process). This is true, but only up to a point: in fact, we see that the SAEs with the least live features tend to correspond to the underlying models that perform the worst, but so do the SAEs with the most live features, while autoencoders with a moderate number of live features tend to correspond to the best underlying models. Interestingly, the SAEs of the best-performing DQNs tend to have significantly fewer features than neurons in those DQNs' hidden layer, despite modeling that hidden layer extremely well and in a manner designed to promote sparsity (using a top-k activation function).

We are able to fit a quadratic regression (the red line in Figure 5) to our live feature counts, and, although live feature counts only explain a modest amount of the total variation in model performance ($R^2 = 0.332$, adjusted $R^2 = 0.268$), they do fit the data at a statistically significant level (using an F-test, we obtain $F = 5.21, p = 0.0146$). This is a surprising and unintuitive result: we are able to predict a nontrivial amount of the variance in DQN performance with a single number by training a SAE. While the predictive value of this method is limited, it is a striking result that a single aggregate statistic is as predictive as it is. This finding may suggest that, given the combination of model, hyperparameters, and task studied in this paper, there could be an optimal, or high-performing set of a certain number of features, such that models lacking these features or containing extraneous features tend to perform worse. However, more study would be needed to conclusively affirm or refute this possibility.

## 5  Conclusion

In this work, sparse autoencoders were trained to find human-interpretable features in deep Q networks trained to play the game Ms. Pacman. Sparse autoencoders consistently found interpretable features in these DQNs, but are generally less effective in this regard than previous work in other domains (e.g. language models) would predict. However, we also find that the quantity of live features present in our sparse autoencoders is a statistically-significant predictor the efficacy of the underlying models. Furthermore, their relationship is not simply linear, with models that have a moderate quantity of live features tending to outperform both very high and very low quantities.

While sparse autoencoders are a promising tool for interpreting and steering the decisions of trained machine learning models, this research suggests that they may also be useful to gauge the overall effectiveness of some models. In particular, by using an objective, numerical standard (number of live features), rather than a subjective standard that requires human interpretation (interpretation of individual features), this technique has the potential to allow for a more rigorous understanding of the behavior of certain classes of machine learning models.

## Acknowledgments and Disclosure of Funding

This research used the Delta advanced computing and data resource which is supported by the National Science Foundation (award OAC 2005572) and the State of Illinois. Delta is a joint effort of the University of Illinois Urbana-Champaign and its National Center for Supercomputing Applications.

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

## A  Individual Features

Two representative interpretable features are discussed below.

Additionally, a full list and visualization of all live features of one of the autoencoders used in this paper can be found at colemanduplessie.github.io/pacman-sae-feature-viewer.

### A.1  Feature 748

Feature 748 activates on 7.97% of all input frames. It consistently activates most strongly when the player is leaving the bottom-right corner after acquiring a power-up (see Figure 6). When activated, its effect is to discourage the model from moving down or to the right[3], and to encourage the model to move in a variety of upward or leftward directions.

---

[3]When the model moves the joystick in an illegal direction (e.g. moving down when already at the bottom of the screen), it has the effect of continuing to move in the current direction.

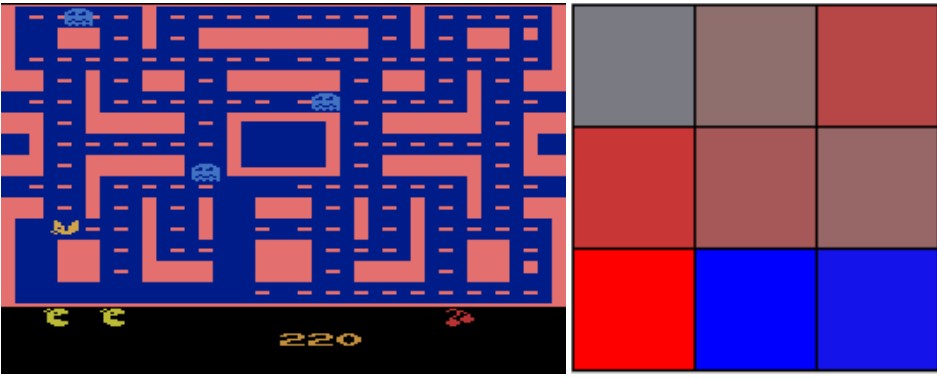

Figure 6: Left: The frame that causes feature 748 to activate most strongly. Right: The effect feature 748 has on the DQN's action selection, where blue represents ablation encouraging the corresponding action (i.e. when the feature is removed, the model values actions denoted with blue more highly).

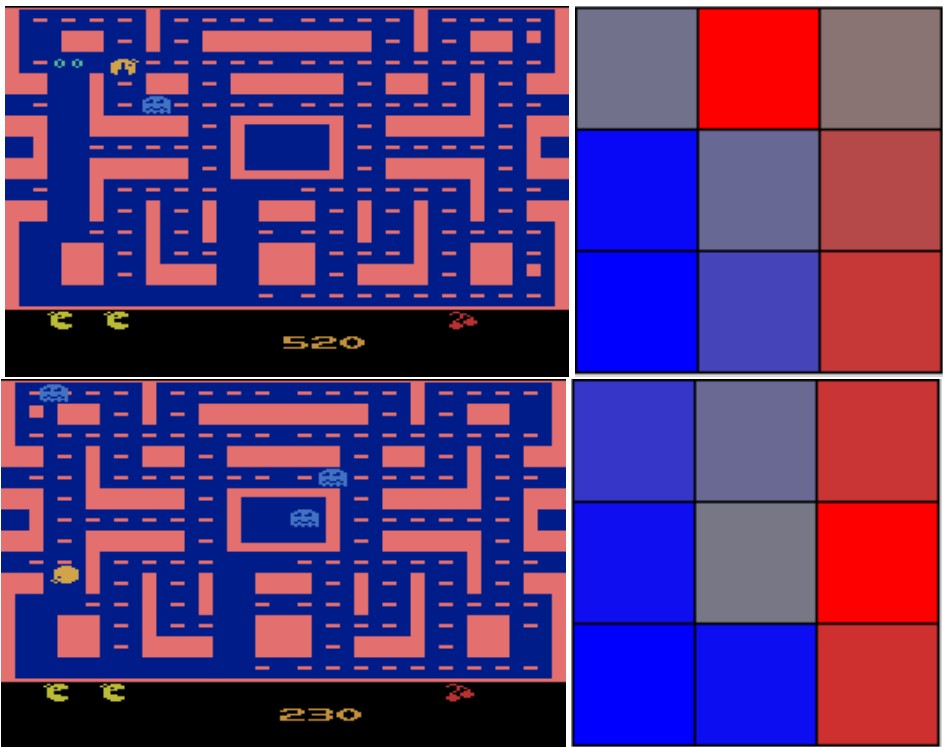

Figure 7: Left: Two representative frames that causes feature 63 to activate strongly. Right: The effects that ablating feature 63 has on the DQN's action selection in the two different cases, where blue represents ablation encouraging the corresponding action (i.e. when the feature is removed, the model values actions denoted with blue more highly).

This is an excellent example of an interpretable feature. It clearly serves one purpose[4], and that purpose seems to align with the model's overall goal of scoring the most points: moving right rather than up is generally not useful, because the model has already scored a significant fraction of the points available to the right.

## A.2 Feature 63

Feature 63 activates on 0.56% of all input frames. It is notable for being bimodal in its activations, appearing to serve a purpose in both cases. In the more common case, depicted at the top of Figure 7, it encourages the model to move up and to the right, continuing toward the power-up in the top-right corner. In the less-common case, depicted at the bottom of the figure, it instead encourages the model to move to the right, discourages it from moving to the left, and weakly discourages it from moving up; this may be beneficial to the model by preventing it from turning left, which allows it to consume the ghost in the upper-right (in Ms. Pacman, the player receives a significant number of points whenever a ghost is consumed, and has a limited time to do so after acquiring a power up).

The fact that this feature is bimodal indicates that the sparse autoencoder has (with respect to this feature) failed: this feature still represents two distinct concepts in superposition, and, if this were not a toy problem or the feature represented significantly more than two concept, the feature would likely be uninterpretable. We would expect this bimodality to be a signal that our sparse autoencoder is too small: that it is impossible to represent all concepts as their own features, so concepts are represented in superposition (see the discussion of feature splitting in Bricken et al. [4]). However, this explanation cannot be true, because our autoencoders contain a very large number of dead features! If it was beneficial to represent the two concepts in this feature as separate features, the model would be able to do so. More research is needed to determine why this feature and similar features remain multi-modal.

---

[4]If it served many different purposes (i.e. represented several concepts in superposition), the set of input states that activate it would be a multi modal distribution, which is not the case.

