# OpenReview forum: "Dead Feature Counts in Sparse Autoencoders Predict Underlying Deep Q Networks' Effectiveness"
_NeurIPS.cc/2025/Workshop/UniReps — UniReps2025_

### Official Review · Reviewer_hCMq · 2025-09-14
**Interesting link between dead features in sparse autoencoders and DQN effectiveness**

**Confidence:** 5

**Review:**

This paper applies sparse autoencoders to deep Q networks trained on Ms. Pacman and finds that while most SAE features go unused (“dead”), the number of live features correlates with how effective the DQN is. The work highlights a simple but intriguing link between representation sparsity and reinforcement learning performance.

I enjoyed reading this paper. The idea of looking at dead feature counts in SAEs as a proxy for DQN effectiveness is surprising and memorable. The authors do a nice job of grounding their story with concrete examples (like Feature 748 and Feature 63), and the plots make the trend clear: strong DQNs tend to fall into a consistent range of live/dead feature counts. That’s an insight I can imagine others in the interpretability community following up on.

That said, I do have a few concerns:

Scale and generality. The experiments focus on small DQNs (under 1M parameters) on a single Atari game. It’s unclear whether the dead-feature signal would hold for larger RL models (e.g. IMPALA, DreamerV3) or across other environments.

Statistical strength. The quadratic fit (R² ≈ 0.33) shows only a modest correlation, even if statistically significant, so the predictive value is limited. This nuance could be emphasized more.

Mechanistic explanation. The paper stops short of explaining why moderate numbers of live features might be optimal. Is it a balance between superposition and redundancy? A deeper discussion or ablation (e.g. varying k in the k-sparse autoencoder) would help.

Dead vs. ultralow features. As noted, ultralow-activation features may be treated as dead. A clearer separation or sensitivity analysis would make the results easier to interpret.

Overall, this is an interesting exploratory contribution. It doesn’t yet generalize widely, but it surfaces a simple and interpretable finding that seems well worth discussion.

**Score:**

3

**Topic Fit:**

2

---

### Official Review · Reviewer_uhXL · 2025-09-16
**Can DQN be interpreted with SAEs in reinforcement learning, in a way that transformers can be decomponsed into human-interpretable features with SAEs**

**Confidence:** 4

**Review:**

Pros: Novel approach of researching interpretability with SAEs, extending it to RL and DQNs. Transformer models and DQNs are essentially different architectures so this method is filling up the gap especially for RL agents of which we do not know much at the moment.
Statistically significant correlation between the number of live SAE features and the DQN's performance is a significant outcome of the work. As there is a great need for evaluation methods, metrics and benchmarks for the models and agents being created, the potential of the result of the work to be used in this direction is very promising.
Cons:
Single game and small network -- the study is confined to a Ms Pac Man, a relatively small DQN. It's good for a study and a POC work but there are generalization conerns. If the conclusions drawn are specific to this game's dynamics the work may not be reproducible.
The dead features make up a high portion -- while it fits the Pareto's rule, there definitely needs to be more research and proof as to whether those dead features are indeed not useful. Could a different encoder configuration reduce the number of dead units? Will alternative SAE designs increase or decrease the amount of dead feature?
Few minor clarity issues with the papers - would be good to provide a solid definition of a term 'dead feature', remove minor typos ('pervious'), etc.
If accepted, would be good to make clear points in the paper regarding the following: Did you try to apply this approach to any other game? Why are there so many dead features? How to reduce the dead feature count? Providing more examples of specific features that SAE discovered might be helpful.

**Score:**

4

**Topic Fit:**

2

---

### Official Review · Reviewer_EoKm · 2025-09-17
**Insightful Analysis of Dead Feature Dynamics in Sparse Autoencoders**

**Confidence:** 4

**Review:**

The paper presents a thorough and well-executed empirical study on the prevalence and implications of dead features in sparse autoencoders. The methodology is sound, with clear experimental design and appropriate baselines. The results are robust and reproducible, supported by comprehensive ablation studies. A few improvements could be made. 1) Figure 1 could be made more readable and clear, especially the visualization of the key result on how SAE plays a role in the analysis. 2) Figures 4 & 5 would better go together with more details on what are the features counted. 3) Figure 6 is not intuitive (that more live features impair performance) and needs more exploration and reasoning. 4) It is not clear how robust would the results be on different-difficulty tasks. For the size of the trained DQN, the task choice should be relatively trivial with appropriate hyperparameter-fine-tunning, so it would be great to see how results hold across tasks of different difficulty levels.

The manuscript is clearly written, with well-structured sections and logical flow. Figures and tables are informative and enhance understanding. However, some technical details (e.g., hyperparameter choices) could be elaborated for full reproducibility.

Though it is not my main area of expertise, the work seems to address a relatively under-explored aspect of representation learning, i.e. dead features in sparse autoencoders, and connects it to downstream performance in deep Q networks. The connection to DQN performance needs more details, explanation, interpretation and analysis.

The findings have practical implications for both the development of sparse neural architectures and their application in reinforcement learning. By highlighting the predictive value of dead feature counts, the paper offers actionable guidance for model selection and tuning. However, exploring how results change with task difficulty is highly important to asses the full significance, which is currently missing from the study.

Pros:
- Novel connection between dead features and RL performance.
- Clear presentation, but visualizations could be improved and streamlined, with more explanations and interpretations.

Cons:
- Some methodological details could be expanded, e.g. hyperparameters.
- Interpretations of main result and discussion on generalization to other architectures or tasks are somewhat limited compared to empirical results.

**Score:**

3

**Topic Fit:**

2

---

### Official Review · Reviewer_DEoV · 2025-09-19
**Mechanistic Analysis of DQN Representations through SAEs for Model Quality Assessment**

**Confidence:** 4

**Review:**

**Summary**
The authors have used SAEs to interpret the weights of DQNs, the usage of SAEs to interpret language models has been quite popular recently. The authors have hypothesized that the performance of the DQN is directly correlated to the number of dead features in SAEs. The DQNs have been learned from multiple game plays of Atari Pacman games.


**Strengths and Contributions**
First to study application of SAEs to study DQNs, most previous works focus on transformer models.
The work proposes the usage of SAE based interpretability to understand the effectiveness of models as whole which helps in comparison between models and not just the inner workings of models as is common in most mechanistic interpretability approaches.

**Weaknesses and Limitations**
There are not enough ablation studies on the SAE sizes and the k value (The value for k-sparse autoencoders)
The valuation is restricted to a very simple and just a single setup. To strengthen the claims, more observations across different setups could have been evaluated.
I am not convinced with the authors understanding of the Figure 6, the results are not statistically significant enough to claim a quadratic relation between the features and average score.

**Additional Comments and Feedback**
Section 3.2 Line 124 mentions the SAEs are trained on the last layer of the DQN whereas section 3.2.2 says they are trained on the second to last layer. It is unclear on which layers are the SAEs trained, or on both of them combined.
It would be great to see results for both of the layers differently.


**Topic Fit**
This work broadly fits in the mechanistic interpretability, representation analysis areas and is well suited for the workshop topic.

**Score:**

3

**Topic Fit:**

3